# Insulin Clearance at the Pubertal Transition in Youth with Obesity and Steatosis Liver Disease

**DOI:** 10.3390/ijms241914963

**Published:** 2023-10-06

**Authors:** Roberto Franceschi, Danilo Fintini, Lucilla Ravà, Michela Mariani, Alessia Aureli, Elena Inzaghi, Stefania Pedicelli, Annalisa Deodati, Carla Bizzarri, Marco Cappa, Stefano Cianfarani, Melania Manco

**Affiliations:** 1Pediatric Department, S. Chiara Hospital of Trento, APSS, 38121 Trento, Italy; roberto.franceschi@apss.tn.it; 2Diabetes and Growth Disorders Unit, Bambino Gesù Children’s Hospital, IRCCS, 00168 Rome, Italy; danilo.fintini@opbg.net (D.F.); michela.mariani@opbg.net (M.M.); alessia.aureli@opbg.net (A.A.); elena.inzaghi@opbg.net (E.I.); stefania.pedicelli@opbg.net (S.P.); annalisa.deodati@opbg.net (A.D.); carla.bizzarri@opbg.net (C.B.); or stefano.cianfarani@uniroma2.it (S.C.); 3Clinical Epidemiology, Bambino Gesù Children’s Hospital, IRCCS, 00168 Rome, Italy; 4Research Unit, Innovative Therapies for Endocrinopathies, Scientific Directorate, Bambino Gesù Children’s Hospital, IRCCS, 00168 Rome, Italy; marco.cappa@opbg.net; 5Department of Systems Medicine, University of Rome ‘Tor Vergata’, 00168 Rome, Italy; 6Department of Women’s and Children’s Health, Karolinska Institutet, 17177 Stockholm, Sweden; 7Research Unit of Predictive and Preventive Medicine, Bambino Gesù Children’s Hospital, 00146 Rome, Italy

**Keywords:** insulin clearance, fatty liver disease, obesity, puberty

## Abstract

No data are available on insulin clearance (Cl_I_) trends during the pubertal transition. The aim of this study was to investigate in 973 youths with obesity whether Cl_I_ in fasting and post-oral glucose challenge (OGTT) conditions varies at the pubertal transition in relation to the severity of obesity and the presence of steatosis liver disease (SLD). The severity of obesity was graded according to the Centers for Disease Control. SLD was graded as absent, mild and severe based on alanine amino transferase levels. Cl_I_ was defined as the molar ratio of fasting C-peptide to insulin and of the areas under the insulin to glucose curves during an OGTT. In total, 35% of participants were prepubertal, 72.6% had obesity class II, and 52.6% had mild SLD. Fasting Cl_I_ (nmol/pmol × 10^−2^) was significantly lower in pubertal [0.11 (0.08–0.14)] than in prepubertal individuals [0.12 (0.09–0.16)] and higher in class III [0.15 (0.11–0.16)] than in class I obesity [0.11 (0.09–0.14)]. OGTT Cl_I_ was higher in boys [0.08 (0.06–0.10)] than in girls [0.07 (0.06–0.09)]; in prepubertal [0.08 (0.06–0.11)] than in pubertal individuals [0.07 (0.05–0.09)]; in class III [0.14 (0.08–0.17)] than in class I obesity [0.07 (0.05–0.10)]; and in severe SLD [0.09 (0.04–0.14)] than in no steatosis [0.06 (0.04–0.17)]. It was lower in participants with prediabetes [0.06 (0.04–0.07)]. OGTT Cl_I_ was lower in youths with obesity at puberty along with insulin sensitivity and greater secretion. The findings suggest that the initial increase in Cl_I_ in youth with severe obesity and SLD is likely to compensate for hyperinsulinemia and its subsequent decrease at the onset of prediabetes and other metabolic abnormalities.

## 1. Introduction

Fasting and post-meal circulating levels of insulin are tightly regulated in humans to maintain euglycemia. Circulating insulin results from the balance between insulin release and clearance [1,2]. Insulin clearance (Cl_I_) refers to the rate at which insulin is removed from the bloodstream Liver, kidney and skeletal muscle, clear plasma insulin, but the liver accounts for up to 80% of the hormone withdrawal during its first-pass transit throughout the portal system [1,2]. This seemingly futile cycle of insulin from the pancreas to the liver is key to prevent the adverse metabolic effects of chronic hyperinsulinemia [3,4]. The liver serves as a gateway for insulin, carrying only the proper amount of the hormone to the periphery in proportion to metabolic need. Insulin that subsists the hepatic first pass reaches the hepatic veins and, thus, the systemic circulation wherein it targets all the tissues [4].

In people living with obesity, who have severe insulin resistance (IR) that is compensated by β-cell insulin hypersecretion, post-prandial Cl_I_ is consequently reduced [5,6,7] and acts as an independent determinant of lower β-cell function over time. Black American adolescents who have more severe IR than white non-Hispanic adolescents show, indeed, reduced insulin clearance [8]. Thus, IR, insulin secretion and clearance are three effectors in an egg–chicken riddle [9]. Reduced Cl_I_ can exacerbate hepatic hyperinsulinemia and IR in a vicious cycle favoring fat accumulation in the parenchyma, which can lead to steatosis liver disease (SLD) [10,11]. However, intrahepatic fat causes hepatic IR and reduced clearance. Therefore, it is not surprising that youths with SLD had lower endogenous insulin clearance and more severe hepatic IR than their peers without SLD [10]. The association between intrahepatic fat and reduced hepatic but not extra-hepatic clearance was more evident in Black American individuals than in Caucasian and Hispanic individuals [10]. Reduced Cl_I_ might explain to some extent the higher rates of prediabetes and type 2 diabetes (T2D) in youths with SLD or in Black American young people independently of whether they have SLD [11]. Reduced Cl_I_ emerges as a key determinant of prediabetes and T2D in adolescents [12,13] who have lower insulin sensitivity, hyper-responsive β-cells, and reduced Cl_I_ compared with adults with altered glucose tolerance [14].

Given the reciprocal tight regulation among the three components of insulin metabolism, it is very likely that Cl_I_ can be differently modulated during the pubertal transition, a life window of great change in the two other components that are insulin sensitivity and secretion [15]. To the best of our knowledge, there are no data on Cl_I_ trends during the pubertal transition in boys and girls with respect to their obesity and SLD condition. The aim of the present study was to investigate whether Cl_I_ varies differently in boys and girls at the pubertal transition and whether the severity of obesity and the presence of SLD affect it.

## 2. Results

Table 1 shows the anthropometrics and metabolic parameters of the 973 non-Hispanic white youths (males 490, 50.4%). A total of 67 (6.9%) participants had prediabetes: 22 (2.3%) had exclusively fasting glucose ≥100 mg/dL; 19 (1.95%) had exclusively 2 h glucose ≥140 mg/dL; 11 (1.1%) had exclusively HbA1 ≥ 5.7%; and 15 (1.5%) had combined prediabetes, i.e., a combination of two of the previous three phenotypes. None had T2D.

In the population (Table 2), we observed significant differences in HOMA-IR (*p* = 0.027), the Matsuda index (*p* = 0.011) and the insulin-to-glucose AUC ratio (*p* = 0.006) between sexes and in OGTT Cl_I_ (*p* = 0.024), which was lower in girls than in boys. Insulin resistance was significantly higher in pubertal individuals than in prepubertal ones (*p* < 0.001), and, conversely, sensitivity was lower (*p* < 0.001). Insulin secretion was increased in the first 30 min (*p* < 0.001) of the OGTT as well as during the whole test (*p* < 0.001) in pubertal individuals as compared to prepubertal ones. Pubertal participants had higher Cl_I_ in the fasting condition (*p* = 0.001) and during the OGTT (*p* = 0.004). Of note, patients with obesity class III had higher fasting and (*p* = 0.001) OGTT Cl_I_ (*p* > 0.001) than patients with class I obesity. Patients with severe SLD had higher OGTT Cl_I_ (*p* < 0.001) than those with no SLD. In patients with prediabetes, fasting Cl_I_ tended to be lower, and OGTT Cl_I_ was significantly higher (*p* = 0.02) than in subjects with normal glucose homeostasis.

Figure 1 shows the Spearman’s correlation matrix of all the continuous variables analyzed.

Table 3 reports the associations of fasting and OGTT Cl_I_ with anthropometric and metabolic variables in the univariate analyses; the strength and direction of each association are depicted in Figure 2.

Fasting Cl_I_ was significantly and positively associated with obesity class III (*p* = 0.001) and grams of fat mass (*p* = 0.005), insulin sensitivity (both the Matsuda index and MISI) and the disposition index (*p* < 0.001 for all of them), and it was negatively associated with age (*p* = 0.001) and puberty (*p* = 0.001), SBP (*p* < 0.0001), waist (*p* = 0.006), HOMA-IR, IGI and the insulin-to-glucose AUC ratio (*p* < 0.001 for all). Cl_I_ following the OGTT was negatively associated with growing age (*p* = 0.006), systolic blood pressure (*p* < 0.001), puberty (*p* = 0.004), HbA1c (*p* < 0.001), HOMA-IR (*p* < 0.001), both parameters of insulin secretion (*p* < 0.001) and prediabetes (*p* = 0.022). It was positively associated with obesity class III and fat mass in grams (*p* < 0.001 for both), severe SLD (*p* < 0.001) and insulin sensitivity (*p* < 0.001 for both parameters).

A multivariate analysis (Table 4) found HOMA-IR as a negative predictor and severe SLD as a positive (*p* < 0.0001 for both) predictor of fasting Cl_I_; puberty (*p* = 0.002), mild and severe SLD (*p* = 0.001 and *p* <0.0001) and total cholesterol (*p* = 0.01) as positive predictors of OGTT clearance; and HDL cholesterol (*p* = 0.002), the triglyceride-to-HDL-cholesterol ratio (*p* = 0.002), LDL cholesterol (*p* = 0.001) and body fat % (*p* = 0.006) as negative predictors in our sample of children and adolescents with obesity.

## 3. Discussion

In the “Bambino” population, we found that fasting Cl_I_ was lower in pubertal than in prepubertal patients with obesity and was upregulated in those with severe obesity.

Likewise, hormone clearance during the OGTT was lower in pubertal participants and in those with prediabetes, while it was upregulated in those with severe SLD and obesity.

Notably, we also found differences in OGTT-derived Cl_I_ between sexes.

The multivariate analysis confirmed the close association of clearance indexes with insulin sensitivity in the positive direction and the parameters of secretion in the negative direction.

Thus, it appears clear that changes in insulin clearance occurring at the pubertal transition add complexity to the comprehension of the different phenotypes of altered glucose homeostasis. Therefore, we believe that insulin clearance deserves assessment in the ward, in addition to sensitivity and secretion, when a young patient with obesity is screened for comorbidities at the pubertal transition, as well as a lab investigation as far as the role that sex and other pivotal hormones may play in the modulation of this under-investigated metabolic path.

Of particular interest is the association of fasting Cl_I_ with MISI, which is a specific estimate of muscle-specific insulin sensitivity and expresses the decay of glucose over time by the average concentration of insulin during the OGTT. Therefore, it is reasonable that it is inversely related to Cl_I_. In this regard, it must be acknowledged that the molar ratio between C-peptide and insulin concentrations does not allow for the dissection of the hepatic and the peripheral contribution to insulin clearance [16], and it resembles more an index of appearance rather than a rate of disappearance of the hormone.

### 3.1. The Role of Puberty

Insulin sensitivity decreases along with pubertal progression, while there is a compensatory increase in insulin secretion. Studies found that Cl_I_ is lower in youth with obesity than in their normal-weight peers [5,7,17]. These studies did not explore Cl_I_ trends during the pubertal transition. In our sample of youths with obesity, we found differences in insulin clearance in the fasting condition and during the OGTT between prepubertal and pubertal individuals who had lower clearance. This association was expected, with it being related to pubertal changes in hormone sensitivity and secretion. However, the association of OGTT Cl_I_ with puberty became positive in the multivariate analysis, probably because of the overarching effect of SLD and severe obesity, whose prevalence is higher in pubertal individuals than in prepubertal ones. We speculate that fasting and post-meal Cl_I_ levels decline with pubertal progression as a consequence of IR and insulin hypersecretion (graphical abstract). Insulin concentrations regulates the rate and the extent of insulin receptor endocytosis since high concentrations of the hormone cause faster internalization of the insulin receptors [18].

Galderisi et al. found that in youth with obesity, reduced insulin sensitivity more than severity of obesity was paralleled by a gradual decrease of hepatic and extrahepatic Cl_I_, as measured by a hyperinsulinemic-euglycemic and a hyperglicemic clamp [11]. The same authors stated that hepatic Cl_I_ plays a major role in the decline of β-cell function [11], contributing to the progression to T2D [11,19], and that Cl_I_ could probably be an early marker of the disease [11,19].

The RISE (Restoring Insulin Sensitivity) study compared the parameters of insulin metabolism in youth and adulthood, investigating Cl_I_ via the fasting C-peptide-to-insulin molar ratio [20]. After adjusting for insulin sensitivity, the authors found that insulin secretion was significantly higher and that Cl_I_ was lower in young individuals than in adult individuals. They speculated that hepatic insulin extraction is reduced in young people to limit the stress on β-cell function in a condition of such increased insulin resistance as during the pubertal transition.

### 3.2. The Role of SLD

In our large sample, OGTT Cl_I_ was upregulated in people with SLD, suspected on the basis of ALT levels. This might occur initially in the natural history of the disease, likely in an attempt to reduce the burden of hepatic hyperinsulinemia that, in turn, exacerbates fat accretion in the liver. Nonetheless, reduced hepatic insulin clearance leads to hyperinsulinemia and, in turn, to hepatic insulin resistance and fat accumulation within the parenchyma, as postulated by Bergman et al. [3]. Previous studies in pediatric SLD reported that the accumulation of intrahepatic fat, quantified using magnetic resonance imaging, drives hepatic gluconeogenesis and IR [10,21,22,23], and children with SLD have a two- to three-fold higher risk of prediabetes and T2D than controls [24]. Our finding is different from that reported in obese children [10,23] and adults with SLD [25,26,27] that showed reduced insulin clearance estimated with the same method that we used in this study. The severity of hyperinsulinemia might be the key to explaining the different findings. At a very high concentration of the hormone, i.e., >500 μU/mL, hepatic insulin receptors are saturated [17]. The saturation of insulin receptors with an increasing pre-hepatic insulin concentration can lead to reduced Cl_I_ and, in turn, to impaired β-cell function [10]. Alternatively, the severity of the hepatic injury can be the key to explaining the divergent results. Hepatocyte apoptosis and hepatic stellate cell activation, which occur with SLD progression, have been suggested to alter the distribution of insulin receptors that mediate insulin clearance in the liver with increased clearance in the presence of hepatic fibrosis [28].

### 3.3. Obesity, Prediabetes and Other Metabolic Abnormalities

Our study demonstrates that, in the absence of overt diabetes, in youth with obesity, OGTT Cl_I_ increases with obesity severity class. This might represent an early mechanism to counterbalance hyperinsulinemia. With insulin receptor saturation, clearance starts decreasing, hence alleviating the stress on β-cells. Theoretically, insulin-resistant individuals can maintain glucose homeostasis through an increase in the insulin secretion rate (AUC insulin over glucose) that, in turn, should determine the increase in Cl_I_ via a physiological mechanism [29]. Our findings confirm that Cl_I_ is a key risk factor for the progression of youths with obesity to prediabetes and T2D. In our series, prediabetes was negatively associated with OGTT Cl_I_, notwithstanding that participants belonging to our sample had no overt T2D and a relatively low rate of impaired fasting glucose and glucose intolerance. Therefore, given the inverse association between clearance during OGTT and prediabetes, we speculate that reduced clearance is a key factor in the pathogenesis and occurrence of prediabetes progressing toward overt diabetes. In adults with T2D, both fasting and post-absorptive Cl_I_ were reduced [19]. Two possible mechanisms have been put forward to explain the association of clearance with diabetes risk: insulin resistance drives reduced clearance and, therefore, leads to diabetes risk; alternatively, reduced insulin degradation resulting from genetics or the environment is a primary factor causing systemic hyperinsulinemia, insulin resistance, β-cell stress and, finally, diabetes [19,30].

In our study, waist circumference, systolic blood pressure and high gamma glutamyl transferase levels, which, indeed, are markers of metabolic syndrome, were associated with reduced Cl_I_ (fasting and OGGT clearance for pressure, fasting for waist and OGTT derived for liver enzymes). This result agrees with the data reported by Pivovarova et al. [29], who demonstrated that OGTT-derived indexes of hepatic Cl_I_ were lower in adults with metabolic syndrome and correlated with estimates of insulin secretion and insulin sensitivity [29]. We confirmed this finding in children and adolescents with obesity.

### 3.4. Strengths and Limitations

This study has some strengths. The sample size was large and homogenous, as all the participants were Italian and evaluated using the same investigative protocol. Our analysis included a complete evaluation of insulin sensitivity, secretion and clearance according to OGTT data, and we evaluated these data along with pubertal progression. As for other studies [31], the method used to estimate insulin clearance has been validated against a clamp-derived method in young individuals with obesity [10], but it does not distinguish peripheral and hepatic contribution to clearance [16]. The hepatic insulin extraction as the C-peptide-to-insulin molar ratio has pitfalls mainly based on the erroneous assumption that C-peptide and insulin have identical kinetics, while they have very different half-lives of 4 vs. 30 min, respectively; C-peptide has mono-compartmental kinetics, and insulin has three compartment kinetics [16]. Furthermore, we derived estimates of sensitivity, secretion and clearance from the same OGTT, and this might have inflated the extent of the association between these parameters. This study also has limitations in that SLD was diagnosed based on elevated liver enzyme levels according to the criteria suggested by Schwimmer et al. [32] and that a liver ultrasound was performed in only 300 patients with no data from liver biopsy, which remains the gold standard of diagnosis.

### 3.5. Conclusions

In adolescents with obesity, post-meal Cl_I_ declines during puberty transition along with reduced insulin sensitivity and increased insulin secretion. With regard to obesity and fatty liver, we postulate that insulin clearance increases first to compensate for the high concentration of circulating insulin; then, it declines when insulin levels become too high, and obesity comorbidities and altered glucose homeostasis manifest. We also agree that reduced clearance contributes robustly to the onset and progression of T2D, as put forward in Najjar, Caprio and Gastaldelli [19]. In this frame, the estimation of insulin clearance could be used as additional data for the early identification of subjects at high risk of metabolic syndrome and T2D. A study of a large multiethnic cohort of children and adolescents with deeper phenotyping is warranted to conclude the role of insulin clearance in the onset and progression of SLD and T2D in youth.

## 4. Materials and Methods

Participants belonged to the “Bambino” meta-cohort [33,34], having been enrolled at the Bambino Gesù Children’s Hospital (OPBG, Ospedale Pediatrico Bambino Gesù), a tertiary referral center for pediatric obesity in Italy. Briefly, this population includes data of children and adolescents who were normal weight (N = 937) or presented with overweight or obesity (N = 3950) at enrolment. Primary care practitioners referred them all from the metropolitan area of Rome. Participants underwent measurements of body weight, height, blood pressure, and biochemistry; a physical examination; and an oral glucose tolerance test (OGTT).

### 4.1. Study Design and Sample Size

For the purposes of this study, we evaluated data of 973 individuals with obesity, aged 3 to 17.9 years old, who had a complete dataset of anthropometrics, Tanner stage, liver function tests and 5-point OGTT glucose, insulin and C-peptide data. Participants’ medical history was recalled by the hospital electronic medical record.

### 4.2. Anthropometric Measurements and Biochemical Assays

Anthropometry was evaluated and laboratory tests were performed in all the participants according to the customary protocol at OPBG. Participants were asked to refrain from intensive physical activity in the 3 days prior to the study. Weight and height were measured using a standard procedure. Up to 3 blood pressure measurements were taken by a physician after a 5 min rest following a standard protocol, and the mean of the 3 measurements was used for the analysis [16].

After an 8–12 h fast, participants underwent an OGTT (1.75 g/kg body weight up to a maximum of 75 g) with flavored glucose (Glucosio Sclavo Diagnostics, 75 g/150 mL, Sovicille, Italy). Blood glucose was assessed using the glucose oxidase technique (Cobas Integra, Roche, Indianapolis, IN, US); insulin and C-peptide levels were assessed using chemiluminescence on an ADVIA Centaur analyzer (Bayer Diagnostics, Siemens Healthcare, Erlagen, Germany; C-peptide intra- and inter-assay coefficients of variation 3.7–4.1 and 1.0–3.3%, respectively). HbA1c was tested using high-performance liquid chromatography (VARIANT II TURBO Hemoglobin Testing System; Bio Rad, Hercules, CA, USA).

### 4.3. Case Definition and Calculations

We calculated body mass index (BMI), and the sex- and age-specific standard deviation scores (SDSs) of BMI [33], grading the severity of obesity as recommended by the Centers for Disease Control. Class I obesity was defined as BMI ≥ 95th percentile, class II as BMI ≥ 120% of the 95th percentile and class III as BMI ≥ 140% of the 95th percentile [33]. Puberty stage was clinically evaluated based on secondary sex physical appearance, i.e., breast shape and the quantity and pattern of pubic hair for girls, and genital development and the quantity and pattern of pubic hair for boys. Puberty was staged according to Tanner [35]. High levels of alanine amino transferases (ALTs) were defined as ALT ≥ 26UI/L in boys and ≥22 UI/L in girls. High ALT levels were used as a surrogate for SLD [32] that was graded as absent, mild (ALT ≥ 26UI/L in boys and ≥22 UI/L in girls) and severe (ALT > 50 UI/L) [32].

The following parameters were calculated:

(1) Insulin resistance/sensitivity: the homeostasis model assessments of fasting insulin resistance (HOMA IR) = fasting insulin (μIU/mL) × fasting glucose (mmol/mL)/22.5 [36]; the Matsuda index = 10,000/√ [(fasting insulin [pmol/L] × fasting glucose [mmol/L]) × (mean OGTT insulin [pmol/L]) × (mean OGTT glucose [mmol/L])] [37]; and the muscle insulin sensitivity index (MISI) as = (dG/dt)/mean plasma insulin concentration, where dG/dt is the rate of decay of the plasma glucose concentration from its peak value to its nadir during the OGTT [38].

(2) Insulin secretion was estimated using the insulinogenic index (IGI) with IGI = [30 min insulin—fasting insulin (pmol/L)]/[30 min glucose—fasting glucose (mmol/L)] [39], and using the ratio of the insulin area under the curve (AUC_I_) to the glucose area under the curve (AUC_G_).

(3) The disposition index (DI) was calculated as follows: DI = ISI × [AUC_30 min insulin_/AUC_30 min glucose_], where AUC_30 min_ is the AUC between baseline and 30 min of the OGTT for insulin (pmol/L) and glucose (mmol/L) measurements [40].

(4) Fasting hepatic Cl_I_ was defined as the ratio of fasting C-peptide to fasting insulin [14]. Because of the equimolar secretion of both peptides and the lack of C-peptide extraction by the liver, under steady-state conditions, the C-peptide-to-insulin molar ratio is proportional to the hepatic clearance rate of insulin (HIC), as reported in other studies [25,29]. The OGTT-derived hepatic Cl_I_ was determined as a ratio of the incremental areas under the curve (AUC) of OGTT (AUC_C-peptide 0–120 min_/AUC_insulin 0–120 min_) [29], and it was previously found to be strongly correlated with metabolic Cl_I_ determined in hyperinsulinemic–euglycemic clamp experiments [25].

In all cases, AUC was calculated using the trapezoidal method.

### 4.4. Statistical Analysis

Data are presented as count and percentage or median and interquartile range (IQR). Data normality was tested using the D’Agostino Pearson test. All the variables had a non-normal distribution, except for diastolic blood pressure. Spearman’s correlation coefficient was used to explore the correlation between continuous variables. Univariable and multivariable quantile regression analyses were performed to evaluate the association between the median values of insulin clearance (fasting and OGTT Cl_I_) and possible determinants and/or confounders. Variables with a *p*-value ≤ 0.1 in the univariable regression were included in the multivariable models. Best fitting models were obtained through a backward selection approach. The results were considered statistically significant at a *p*-value < 0.05. Statistical analysis was performed by using Stata 17.1 software.

## Figures and Tables

**Figure 1 ijms-24-14963-f001:**
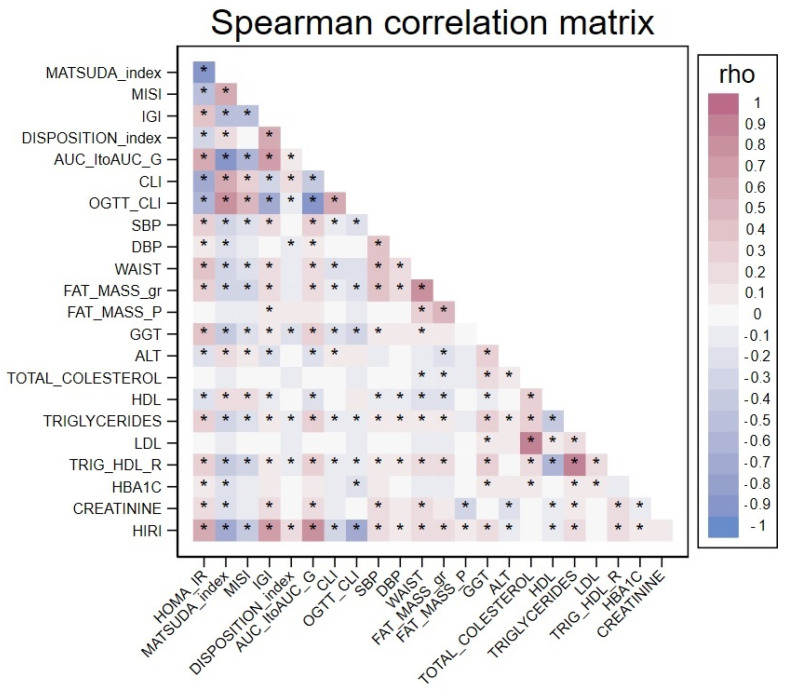
Spearman’s correlation matrix. The colours represent the Spearman correlation coefficients between each continuous variable, the * indicate statistically significant coefficients (*p* < 0.005).

**Figure 2 ijms-24-14963-f002:**
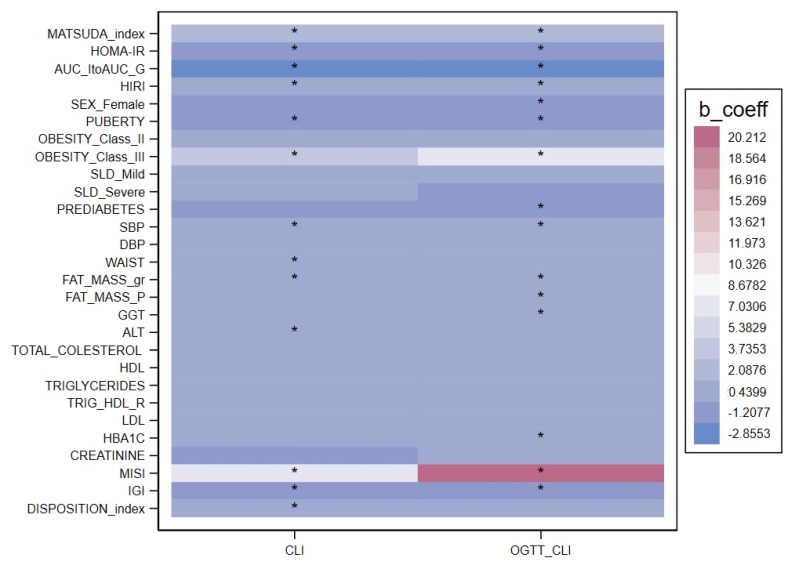
Heat map of univariable quantile regression for fasting and OGTT insulin clearance. The colors represent coefficients of univariable quantile regressions for each of the clearance (CLI, OGTT_CLI), the * indicate the statistically significant coefficients (*p* < 0.005).

**Table 1 ijms-24-14963-t001:** Anthropometric and biochemical characteristics of the 973 children and adolescents with obesity.

Characteristics		Median (IQR) or N (%)
Sex (male/female)		490 (50.4%)/483 (49.6%)
Age (years)		12.0 (10.0–13.8)
BMI (Kg/m^2^)		29.4 (26.8–32.8)
BMI z-score (SDS)		2.3 (2.0–2.6)
Waist circumference (cm)		97.0 (89.5–105.5)
Systolic blood pressure (mmHg)		117 (110–125)
Diastolic blood pressure (mmHg)		66 (60–73)
Fat mass (Kg)		27.9 (22.5–33.7)
Fat mass (%)		40.8 (37.8–43.8)
Prepubertal_Tanner	Stage I	343 (35.2%)
Puberty		630 (64.8%)
Early puberty_Tanner	Stage II	40 (6.3%)
	Stage III	170 (27.0%)
	Stage IV	208 (33.1%)
	Stage V	212 (33.6%)
ALT (IU/L)		28 ± 14
GGT (IU/L)		16 ± 11
Total cholesterol (mg/dL)		156 (139–175)
HDL (mg/dL)		45 (39–52)
LDL (mg/dL)		92 (76–107)
Triglycerides (mg/dL)		80 (59–112)
Triglyceride-to-HDL-cholesterol ratio		1.8 (1.2–2.7)
HbA1c (mmol/L)		34 (32–36)
Fasting glucose (mmol/L)		4.6 (4.3–4.9)
Fasting insulin (pmol/L)		17 0 (11.5–23.7)
Fasting C-peptide (ng/mL)		1.9 (1.4–2.5)
2 h glucose (mmol/L)		6.1 (5.4–6.9)
HOMA IR		3.5 (2.5–5.0)
HIRI		716 (397–1118)
Insulinogenic index (pmol/µmol)		2.1 (1.3–3.4)
Matsuda index (µmol/kg/pM)		2.6 (1.8–3.9)
MISI (µmol/kg/pM)		0.08 (0.04–0.16)
Cl_I_ (nmol/pmol)		0.11 (0.09–0.14)
OGTT Cl_I_ (nmol/pmol)		0.07 (0.06–0.10)
AUC insulin to AUC glucose (pmol/mmol)		0.85 (0.57–1.26)
Disposition index		5.4 (3.4–8.2)

Data are shown as median and interquartile range (IQR) or number and (percentage). Abbreviations: alanine amino transferase, ALT; body mass index, BMI; area under the curve, AUC; gamma glutamyl transferase, GGT; hemoglobin A1c, HbA1c; hepatic insulin resistance index, (HIRI); homeostasis model assessment of insulin resistance, HOMA-IR; muscle insulin sensitivity, (MISI); insulinogenic index, IGI; oral glucose tolerance test, OGTT.

**Table 2 ijms-24-14963-t002:** Univariate analysis of insulin sensitivity, secretion and clearance parameters.

	N	HOMA-IR	Matsuda Index (µmol/kg/ pmol)	IGI (pmol/µmol)	*DI*	*AUC_I_/AUC_G_*	*Fasting Cl_I_* (nmol/pmol × 10^−2^)	*OGTT Cl_I_* (nmol/pmol × 10^−2^)
Males	490	3.31 (2.14–4.62)	2.81 (1.90–4.41)	1.90 (1.16–3.18)	5.38 (3.45–8.23)	0.78 (0.51–1.19)	0.11 (0.09–0.15)	0.08 (0.06–0.10)
Females	483	3.64 (2.52–5.49)	2.40 (1.69–3.55)	2.29 (1.37–3.72)	5.29 (3.27–8.14)	0.91 (0.66–1.35)	0.11 (0.09–0.14)	0.07 (0.06–0.09)
*p*-value		0.027	0.011	0.005	0.902	0.006	0.232	0.024
Prepuberty	335	2.51 (1.52–3.72)	3.29 (2.16–5.14)	1.62 (0.93–2.91)	5.42 (3.62–8.56)	0.71 (0.47–1.05)	0.12 (0.09–0.16)	0.08 (0.06–0.11)
Puberty	615	3.90 (2.84–5.73)	2.33 (1.59–3.39)	2.32 (1.50–3.73)	5.30 (3.32–8.08)	0.92 (0.63–1.37)	0.11 (0.08–0.14)	0.07 (0.05–0.09)
*p*-value		<0.001	<0.001	<0.001	0.655	<0.001	0.001	0.004
Obesity class I	262	3.19 (2.09–4.39)	2.92 (1.98–4.26)	2.00 (1.15–3.15)	5.24 (3.49–7.89)	0.79 (0.56–1.16)	0.11 (0.09–0.14)	0.07 (0.05–0.10)
Obesity class II	694	3.57 (2.36–5.11)	2.54 (1.69–3.73)	2.15 1.32–3.49)	5.38 (3.32–8.21)	0.86 (0.58–1.29)	0.11 (0.09–0.14)	0.07 (0.06–0.09)
Obesity class III	17	2.42 (1.69–3.24)	4.72 (2.18–7.60)	1.72 (0.40–2.09)	5.28 (4.17–12.45)	0.41 (0.33–0.70)	0.15 (0.11–0.16)	0.14 (0.08–0.17)
*p*-value		class II vs. I 0.022 class III vs. I 0.270	class II vs. I 0.052 class III vs. I 0.006	class II vs. I 0.378 class III vs. I 0.632	class II vs. I 0.717 class III vs. I 0.975	class II vs. I 0.197 class III vs. I 0.034	class II vs. I 0.994 class III vs. I 0.001	class II vs. I 0.938 class III vs. I <0.001
No SLD	418	3.99 (2.94–5.74)	2.14 (1.52–3.16)	2.94 (1.61–4.19)	5.91 (3.59–8.52)	1.09 (0.78–1.56)	0.09 (0.08–0.12)	0.06 (0.04–0.07)
Mild SLD	511	3.74 (2.54–5.44)	2.45 (1.65–3.69)	2.64 (1.59–3.92)	5.98 (3.76–9.01)	1.01 (0.65–1.45)	0.10 (0.08–0.13)	0.05 (0.04–0.07)
Severe SLD	44	4.22 (2.35–6.39)	3.40 (1.78–6.49)	2.21 (0.03–3.45)	5.66 (0.16–15.22)	0.69 (0.33–1.50)	0.10 (0.08–0.17)	0.09 (0.04–0.14)
*p*-value		Mild vs. No 0.296 Severe vs. No 0.223	Mild vs. No 0.151 Severe vs. SLD 0.023	Mild vs. No 0.186 Severe vs. No 0.318	Mild vs. No 0.955 Severe vs. No 0.876	Mild vs. No 0.339 Severe vs. No 0.220	Mild vs. No 0.195 Severe vs. No 0.664	Mild vs. No 0.471 Severe vs. No <0.001
Normoglycemia	906	3.35 (2.24–4.71)	2.69 (1.79–4.00)	2.09 (1.25–3.36)	5.40 (3.45–8.27)	0.84 (0.55–1.26)	0.11 (0.09–0.15)	0.07 (0.06–0.10)
Prediabetes	67	5.39 (3.62–8.16)	1.68 (1.16–2.29)	2.45 (1.40–4.68)	4.03 (3.14–6.06)	1.09 (0.79–1.50)	0.10 (0.08–0.13)	0.06 (0.04–0.07)
*p*-value		<0.001	0.005	0.250	0.075	0.016	0.081	0.022

Data are shown as median and interquartile range (IQR). Abbreviations: homeostasis model assessment of insulin resistance, HOMA-IR; insulinogenic index, IGI; disposition index, DI; area under the curve, AUC; insulin clearance, Cl_I_; oral glucose tolerance test, OGTT.

**Table 3 ijms-24-14963-t003:** Association of fasting and OGTT insulin clearance with anthropometric and metabolic parameters.

	Fasting Cl_I_ (nmol/pmol × 10^−2^)	*p*-Value	OGTT Cl_I_ (nmol/pmol × 10^−2^)	*p*-Value
Sex (M/F)	−0.38 (−1.02; 0.25)	0.232	−0.69 (−1.3; −0.09)	0.024
Age (years)	−0.18 (−0.29; −0.08)	0.001	−0.14 (−0.24; −0.04)	0.006
Puberty	−1.14 (−1.80; −0.48)	0.001	−0.86 (−1.44; −0.28)	0.004
Systolic blood pressure (mmHg)	−0.06 (−0.09; −0.03)	<0.001	−0.06 (−0.08; −0.03)	<0.001
Diastolic blood pressure (mmHg)	−0.01 (−0.04; 0.03)	0.754	−0.02 (−0.06; 0.01)	0.161
Waist circumference (cm)	−0.04 (−0.07; −0.01)	0.006	−0.02 (−0.06; 0.01)	0.215
Fat mass (g)	0.00 (0.00–0.00)	0.005	0.00 (0.00–0.00)	<0.001
Fat mass (%)	−0.05 (−0.14; 0.05)	0.344	−0.07 (−0.13; −0.005)	0.035
Obesity class II Obesity class III	0.00 (−0.73; 0.72) 4.21 (1.75; 6.67)	0.994 0.001	0.03 (−0.69; 0.75) 6.31 (3.12; 9.49)	0.938 <0.001
Mild SLD Severe SLD	0.46 (−0.24; 1.15) 0.37 (−1.29; 2.03)	0.195 0.664	−0.34 (−1.28; 0.59) 8.27 (4.47; 12.07)	0.471 <0.001
GGT (IU/L)	−0.02 (−0.06; 0.01)	0.212	−0.04 (−0.07; −0.06)	0.022
Total cholesterol (mg/dL)	−0.01 (−0.02; 0.01)	0.321	−0.00 (0.01; 0.01)	0.678
HDL (mg/dL)	0.00 (−0.03; 0.02)	0.865	0.01 (−0.01; 0.03)	0.418
LDL (mg/dL)	0.00 (−0.02; 0.01)	0.531	−0.00 (−0.01; 0.01)	0.787
Triglycerides (mg/dL)	−0.00 (−0.01; 0.00)	0.460	−0.00 (−0.01; 0.00)	0.124
TGD/HDL	−0.01 (−0.05; 0.03)	0.750	−0.01(−0.04; 0.01)	0.333
Creatinine (mg/dL)	−0.73 (−4.32; −2.9)	0.692	0.58 (−3.43; 4.58)	0.776
HbA1c (mmol/L)	−0.06 (−0.12; 0.01)	0.093	−0.07 (−0.11; −0.04)	<0.001
HOMA IR	−0.79 (−0.88; −0.72)	<0.001	−0.42 (−0.49; −0.33)	<0.001
HIRI	0.00 (0.00; −0.00)	<0.001	0.00 (0.00; −0.00))	<0.001
Matsuda index (µmol/kg/pM)	1.59 (1.45; 1.73)	<0.001	1.28 (1.18; 1.39)	<0.001
IGI (pmol/µmol)	−0.46 (−0.57; −0.36)	<0.001	−0.54 (−0.61; −0.47)	<0.001
MISI (µmol/kg/pM)	7.16 (5.17; 9.16)	<0.001	21.0 (17.70; 24.37)	<0.001
AUC insulin/glucose	−3.30 (−4.0; −2.6)	<0.001	−3.67 (−4.07; −3.28)	<0.001
Disposition index	0.13 (0.08; 0.19)	<0.001	−0.02 (−0.06; −0.02)	0.297
Any prediabetes	1.10 (−2.34; 0.14)	0.081	−1.69 (−3.14; −0.25)	0.022

Data are reported as coefficient,95% CI and *p*-value at the univariate quantile regression. Abbreviations: alanine amino transferase, ALT; body mass index, BMI; area under the curve, AUC; gamma glutamyl transferase, GGT; clearance, Cl; hemoglobin A1c, HbA1c; hepatic insulin resistance index, (HIRI); homeostasis model assessment of insulin resistance, HOMA-IR; muscle insulin sensitivity, (MISI); insulinogenic index, IGI; oral glucose tolerance test, OGTT.

**Table 4 ijms-24-14963-t004:** Parameters of multivariate analysis to predict insulin clearance.

Fasting Cl_I_		*p*-Value
Insulinogenic index (pmol/µmol)	−0.43 (−0.69; −0.17)	<0.001
HDL (mg/dL)	−0.04 (−0.07; −0.02)	<0.001
Disposition index	0.19 (0.08; 0.30)	<0.001
MISI (µmol/kg/pM)	−8.61 (−12.89; −4.32)	<0.001
Matsuda index (µmol/kg/pM)	1.55 (1.35; 1.74)	<0.001
**OGTT-derived Cl_I_**		
HbA1c (mmol/L)	−0.039 (−0.058; −0.020)	<0.001
Matsuda index (µmol/kg/pM)	0.490 (0.266; 0.714)	<0.001
AUC insulin/glucose	−1.172 (−1.645; −0.698)	<0.001

Data are reported as coefficient and (95% CI). Abbreviations: hemoglobin A1c, HbA1c; high-density lipoprotein; HDL; insulin clearance, Cl_I_; MISI, muscle insulin sensitivity index; oral glucose tolerance test, GTT.

## Data Availability

Data are available from the corresponding author upon reasonable request.

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
