# Peer review of "Insulin Clearance at the Pubertal Transition in Youth with Obesity and Steatosis Liver Disease"

_ijms, 2023, doi:10.3390/ijms241914963_

Round 1

Reviewer 1 Report

This communication by Franceschi et al., focuses on the relationship between prediabetes, FLD and insulin clearance in girls and boys during pubertal transition. This is a timely study, as the measurement of insulin clearance during OGTT is gaining traction, despite the fact that it primarily relates to hepatic insulin clearance, in this respect, the authors should make a note of it. Nonetheless, the conclusion is supported by the data presented. The number of subjects recruited into the study is satisfactory (~1000). The subjects pool is as homogeneous as it can get outside lab animals.  The strengths and weaknesses of the study are well spelled out by the authors.  

This reviewer recommends minor revision and emphasis that the approaches taken (fasting and OGTT-related insulin clearance) assess hepatic, but not extra-hepatic insulin clearance, as discussed by Piccinini and Bergman (Diabetes Care  2020). 

Moreover, FLD is now referred to as Steatosis Liver Disease. Hence, it should be abbreviated as (SLD) to be consistent with the current recommendations.

Overall, this reviewer is very enthusiastic for this communication. 

The manuscript is well written. Few statements could be edited to make them more attractive and catchy.  To name a few:

- in the abstract: 5th lane - "basing" should be replaced  with "based".

- the first paragraph of the Discussion should be better clarified and stated. 

Author Response

This communication by Franceschi et al., focuses on the relationship between prediabetes, FLD and insulin clearance in girls and boys during pubertal transition. This is a timely study, as the measurement of insulin clearance during OGTT is gaining traction, despite the fact that it primarily relates to hepatic insulin clearance, in this respect, the authors should make a note of it. Nonetheless, the conclusion is supported by the data presented. The number of subjects recruited into the study is satisfactory (~1000). The subjects pool is as homogeneous as it can get outside lab animals.  The strengths and weaknesses of the study are well spelled out by the authors.  

This reviewer recommends minor revision and emphasis that the approaches taken (fasting and OGTT-related insulin clearance) assess hepatic, but not extra-hepatic insulin clearance, as discussed by Piccinini and Bergman (Diabetes Care  2020). 

This concept was clearly stated in section “3.4. Strengths and limitations” (see line 241). To emphasise we added a sentence in the beginning of the revised discussion (see line 160): From the text: … To this regard, it must be acknowledged that the molar ratio between c-peptide and insulin concentrations does not allow dissecting the hepatic and the peripheral contribution to the insulin clearance [26]…

Moreover, FLD is now referred to as Steatosis Liver Disease. Hence, it should be abbreviated as (SLD) to be consistent with the current recommendations.

FLD has been replaced by SLD in the title and whole text.

Overall, this reviewer is very enthusiastic for this communication. 

We thank the reviewer and we are glad of his/her enthusiasm.

Comments on the Quality of English Language

The manuscript is well written. Few statements could be edited to make them more attractive and catchy.  To name a few:

- in the abstract: 5th lane - "basing" should be replaced  with "based".

Changed accordingly.

- the first paragraph of the Discussion should be better clarified and stated. 

The whole text has been edited to improve language and make it catchier.

Reviewer 2 Report

This study is highly intriguing. The results of this research indicate that the initial increase in ClI in adolescents with severe obesity and FLD may potentially compensate for hyperinsulinemia, followed by a subsequent decrease in prediabetes and other metabolic abnormalities. I find this to be a fascinating discovery.

However, there are certain aspects of the manuscript that require revision.

1.First and foremost, the clinical implications of this study need to be underscored in the "Discussion" section. Based on the findings, we could identify an early increase in Fasting Cl or OGTT CI in adolescent patients, allowing for early screening. However, the influence of hormones during adolescence on lipid metabolism and insulin sensitivity needs further explanation. It is crucial to elaborate on why the authors specifically analyzed "adolescent patients" to highlight the significance of the conclusions.

2.Additionally, when presenting the research results, it might be more reader-friendly and informative to use Mean (Median) instead of Mean (SD). This change could enhance the readers' comprehension of the actual distribution of data in the study population.

3.Furthermore, did the authors confirm if their study population met the assumptions for normality in the analyses? In this study, I believe that using generalized estimating equations (GEE) may be more suitable than Univariable and multivariable linear regression analyses.

4.Moreover, did the authors conduct Pearson correlation analyses on the data before data mining? If not, I recommend conducting these analyses initially to confirm any existing correlations among the data variables before proceeding with regression analyses.

Incorporating these revisions and clarifications into the manuscript should enhance its overall quality and understanding for readers.

Author Response

Reviewer #2

This study is highly intriguing. The results of this research indicate that the initial increase in ClI in adolescents with severe obesity and FLD may potentially compensate for hyperinsulinemia, followed by a subsequent decrease in prediabetes and other metabolic abnormalities. I find this to be a fascinating discovery.

Thanks for your appreciation of our work.

However, there are certain aspects of the manuscript that require revision.

1.First and foremost, the clinical implications of this study need to be underscored in the "Discussion" section. Based on the findings, we could identify an early increase in Fasting Cl or OGTT CI in adolescent patients, allowing for early screening. However, the influence of hormones during adolescence on lipid metabolism and insulin sensitivity needs further explanation. It is crucial to elaborate on why the authors specifically analyzed "adolescent patients" to highlight the significance of the conclusions.

We highlight in the early discussion that clearance must have a place in the clinical evaluation of the adolescent at the pubertal transition, while the interplay between sex and puberty linked hormones deserves investigation. See lines 151-156

  1. Additionally, when presenting the research results, it might be more reader-friendly and informative to use Mean (Median) instead of Mean (SD). This change could enhance the readers' comprehension of the actual distribution of data in the study population.

Done. Data are now presented as median and interquartile range and all the tables have been reviewed accordingly. Models are now presented on median distributions.

Reviewer #2

This study is highly intriguing. The results of this research indicate that the initial increase in ClI in adolescents with severe obesity and FLD may potentially compensate for hyperinsulinemia, followed by a subsequent decrease in prediabetes and other metabolic abnormalities. I find this to be a fascinating discovery.

Thanks for your appreciation of our work.

However, there are certain aspects of the manuscript that require revision.

1.First and foremost, the clinical implications of this study need to be underscored in the "Discussion" section. Based on the findings, we could identify an early increase in Fasting Cl or OGTT CI in adolescent patients, allowing for early screening. However, the influence of hormones during adolescence on lipid metabolism and insulin sensitivity needs further explanation. It is crucial to elaborate on why the authors specifically analyzed "adolescent patients" to highlight the significance of the conclusions.

We highlight in the early discussion that clearance must have a place in the clinical evaluation of the adolescent at the pubertal transition, while the interplay between sex and puberty linked hormones deserves investigation. See lines 151-156

  1. Additionally, when presenting the research results, it might be more reader-friendly and informative to use Mean (Median) instead of Mean (SD). This change could enhance the readers' comprehension of the actual distribution of data in the study population.

Done. Data are now presented as median and interquartile range and all the tables have been reviewed accordingly. Models are now presented on median distributions.

The authors present a cross-sectional study on the association of age/pubertal stage and insulin clearance in children and adolescents with obesity.

The overall rationale is clear.

Abstract:

Line 20: ClI, not CII

Edited.

Methods section of the abstract is too short.

We added detail on grading of obesity and calculation of clearance during the OGTT.

Introduction:

L.48-62: Please specify ClI for fasting or postprandial version.

Post prandial.

Methods:

Please clarify, if you checked for normal distribution of your data.

Data were checked for distribution as described in the statistical analysis section and variables with skewed distribution listed.

LIver fat assessment by a single transaminase is not ideal. Please check here, if a suitable liver fat index can be calculated from your data. (https://pubmed.ncbi.nlm.nih.gov/37577225/)

We agree with the comment on diagnostic accuracy of transaminase levels. With regard to surrogate indexes we read with interest the suggested paper by Reinshagen M et al. The authors in the manuscript resume a number of surrogate estimates of liver fat. Surrogate indexes reported have been all developed in adult populations as the authors highlighted. Some of them are based on clinical variables such as waist and triglycerides (i.e. the FLI by Bedogni et al). In the past we tried validating clinical variables-based indexes of steatosis in a population of about 200 individuals_ some of them belonging to the present population_ against liver biopsy. Accuracy of such indexes was not satisfactory. Findings of the validation study of several indexes in a paediatric population by Koot BG, et al. are in agreement with findings of our attempts (Accuracy of prediction scores and novel biomarkers for predicting nonalcoholic fatty liver disease in obese children. Obesity (Silver Spring) 2013;21(3):583–590. doi: 10.1002/oby.20173.) The Koot et al ‘study has been also quoted by Reinshagen M et al who write: “Koot et al. found that these scores are poor predictors of NAFLD in obese children. The cohort (119 severely obese children (14.3±2.1 years of age, BMI z-score 3.35±0.35; 47% NAFLD cases) was investigated with MR spectroscopy as gold standard; FLI, NAFLD-LFS, and HSI were assessed. As these scores were developed for adult populations, their poor performance in paediatric patients is not entirely unexpected. ….”. Another argument to use transaminases instead of scores is that the most of the scores are based on metabolic variables (i.e. waist) that per sé are associated with insulin clearance and this can inflate any eventual association. Other scores are based on variables that are not evaluated routinely in the ward.

We acknowledge the use of liver function test s as surrogate of steatosis in the discussion. See lines 298-301.

Results:

Please re-check sex ratio; numbers in text and table are discordant.

We apologise for typos. We have corrected them.

Please define "combined prediabetes".

We explained in the text: “i.e. the combination of two of the previous three phenotypes”

Please use adequate decimals; BMI, weights and WC with one decimal, RR, liver enzymes, lipids ... without. Raw data precision is the standard for that decision.

Done

Obesity classes do not add up to n=973.

See above.

NAFLD classes are mis-aligned and do not add up to n=973.

See above.

Why did you calculate the MISI (Abdul-Ghani et al.), but not HIRI?

HIRI has been calculated and median values added. Having used HOMA-IR as rough index of hepatic insulin resistance, we did not mention HIRI in the previous version of the study. Both MISI and HIRI are not reported in table 2 to avoid redundancy of information. However they are reported in tables 1, 3 and were used to compute models in table 4.

Please use "<0,001" instead of "0,000".

Done.

Please use three-digit p values, consistently.

Done.

Table 2: Inflated data analysis; correction for multiple testing required. What is the rationale for a correlation between creatinine and ClI?

Table 2 has been renumbered to 3 in the present draft. It resumes simple univariate analyses. We run univariate analyses for all the variables including creatinine. No clue about an association of creatinine levels and clearance though creatinine can be regarded as indirect index of muscle mass. Then, correction for all the possible confounders was considered for table 4 (Multivariable analyses).

Table 2: Disposition index. The confidence interval implies significance. Please re-check.

Done.

Table 3: MISI obesity class III seems to have an extreme outlier. Please check.

Done.

Table 3: Some given p values are not significant, even though they indicate trends. Please omit them.

Done.

Discussion:

CLI shows a positive association with obesity and NAFLD, but an inverse association with prediabetes. Please clarify.

We clarify that the positive association occurs unless and until prediabetes or even overt diabetes occurs. Please see section 3.3 Obesity, prediabetes and other metabolic abnormalities and particularly lines 222-224. From the discussion: “Therefore, given the inverse association between clearance during OGTT and prediabetes, we speculate that reduced clearance is key factor in the pathogenesis and occurrence of prediabetes progressing toward overt diabetes.”

Line 206: Pivovarova

Edited.

Further evaluation of the discussion depends on major revision of the aforementioned points.

We believe you will be satisfied reading the revised draft. Otherwise, we will be glad to address further points that may arise.

Overall: The terms "increase" or "decrease" describe processes, changes. In this cross-sectional study, no such changes can be described.

We edited accordingly the text.

Comments on the Quality of English Language few changes needed.

The whole text has been edited.

3.Furthermore, did the authors confirm if their study population met the assumptions for normality in the analyses? In this study, I believe that using generalized estimating equations (GEE) may be more suitable than Univariable and multivariable linear regression analyses.

We agree linear regression was not the best choice. Despite the quite large sample size, there was lack of normality in most of the variables. Hence, we used quantile regression around median instead.

4.Moreover, did the authors conduct Pearson correlation analyses on the data before data mining? If not, I recommend conducting these analyses initially to confirm any existing correlations among the data variables before proceeding with regression analyses.

Done. We enclose in the present draft figure 1 showing matrix of correlations.

Incorporating these revisions and clarifications into the manuscript should enhance its overall quality and understanding for readers.

We agree that comments were very useful to improve quality of the manuscript and we are grateful to this reviewer.

Reviewer 3 Report

The authors present a cross-sectional study on the association of age/pubertal stage and insulin clearance in children and adolescents with obesity.

The overall rationale is clear.

Abstract:
Line 20: ClI, not CII
Methods section of the abstract is too short.

Introduction:

L.48-62: Please specify ClI for fasting or postprandial version.

Methods:

Please clarify, if you checked for normal distribution of your data.

LIver fat assessment by a single transaminase is not ideal.

Please check here, if a suitable liver fat index can be calculated from your data. (https://pubmed.ncbi.nlm.nih.gov/37577225/)

Results:

Please re-check sex ratio; numbers in text and table are discordant.

Please define "combined prediabetes".

Please use adequate decimals; BMI, weights and WC with one decimal, RR, liver enzymes, lipids ... without. Raw data precision is the standard for that decision.

Obesity classes do not add up to n=973.

NAFLD classes are mis-aligned and do not add up to n=973.

Why did you calculate the MISI (Abdul-Ghani et al.), but not HIRI?

Please use "<0,001" instead of "0,000".

Please use three-digit p values, consistently.

Table 2: Inflated data analysis; correction for multiple testing required. What is the rationale for a correlation between creatinine and ClI?

Table 2: Disposition index. The confidence interval implies significance. Please re-check.

Table 3: MISI obesity class III seems to have an extreme outlier. Please check.

Table 3: Some given p values are not significant, even though they indicate trends. Please omit them.

Discussion:

CLI shows a positive association with obesity and NAFLD, but an inverse association with prediabetes. Please clarify.

Line 206: Pivovarova

Further evaluation of the discussion depends on major revision of the aforementioned points.

Overall: The terms "increase" or "decrease" describe processes, changes. In this cross-sectional study, no such changes can be described.

few changes needed

Author Response

The authors present a cross-sectional study on the association of age/pubertal stage and insulin clearance in children and adolescents with obesity.

The overall rationale is clear.

Abstract:

Line 20: ClI, not CII

Edited.

Methods section of the abstract is too short.

We added detail on grading of obesity and calculation of clearance during the OGTT.

Introduction:

L.48-62: Please specify ClI for fasting or postprandial version.

Post prandial.

Methods:

Please clarify, if you checked for normal distribution of your data.

Data were checked for distribution as described in the statistical analysis section and variables with skewed distribution listed.

LIver fat assessment by a single transaminase is not ideal. Please check here, if a suitable liver fat index can be calculated from your data. (https://pubmed.ncbi.nlm.nih.gov/37577225/)

We agree with the comment on diagnostic accuracy of transaminase levels. With regard to surrogate indexes we read with interest the suggested paper by Reinshagen M et al. The authors in the manuscript resume a number of surrogate estimates of liver fat. Surrogate indexes reported have been all developed in adult populations as the authors highlighted. Some of them are based on clinical variables such as waist and triglycerides (i.e. the FLI by Bedogni et al). In the past we tried validating clinical variables-based indexes of steatosis in a population of about 200 individuals_ some of them belonging to the present population_ against liver biopsy. Accuracy of such indexes was not satisfactory. Findings of the validation study of several indexes in a paediatric population by Koot BG, et al. are in agreement with findings of our attempts (Accuracy of prediction scores and novel biomarkers for predicting nonalcoholic fatty liver disease in obese children. Obesity (Silver Spring) 2013;21(3):583–590. doi: 10.1002/oby.20173.) The Koot et al ‘study has been also quoted by Reinshagen M et al who write: “Koot et al. found that these scores are poor predictors of NAFLD in obese children. The cohort (119 severely obese children (14.3±2.1 years of age, BMI z-score 3.35±0.35; 47% NAFLD cases) was investigated with MR spectroscopy as gold standard; FLI, NAFLD-LFS, and HSI were assessed. As these scores were developed for adult populations, their poor performance in paediatric patients is not entirely unexpected. ….”. Another argument to use transaminases instead of scores is that the most of the scores are based on metabolic variables (i.e. waist) that per sé are associated with insulin clearance and this can inflate any eventual association. Other scores are based on variables that are not evaluated routinely in the ward.

We acknowledge the use of liver function test s as surrogate of steatosis in the discussion. See lines 298-301.

Results:

Please re-check sex ratio; numbers in text and table are discordant.

We apologise for typos. We have corrected them.

Please define "combined prediabetes".

We explained in the text: “i.e. the combination of two of the previous three phenotypes”

Please use adequate decimals; BMI, weights and WC with one decimal, RR, liver enzymes, lipids ... without. Raw data precision is the standard for that decision.

Done

Obesity classes do not add up to n=973.

See above.

NAFLD classes are mis-aligned and do not add up to n=973.

See above.

Why did you calculate the MISI (Abdul-Ghani et al.), but not HIRI?

HIRI has been calculated and median values added. Having used HOMA-IR as rough index of hepatic insulin resistance, we did not mention HIRI in the previous version of the study. Both MISI and HIRI are not reported in table 2 to avoid redundancy of information. However they are reported in tables 1, 3 and were used to compute models in table 4.

Please use "<0,001" instead of "0,000".

Done.

Please use three-digit p values, consistently.

Done.

Table 2: Inflated data analysis; correction for multiple testing required. What is the rationale for a correlation between creatinine and ClI?

Table 2 has been renumbered to 3 in the present draft. It resumes simple univariate analyses. We run univariate analyses for all the variables including creatinine. No clue about an association of creatinine levels and clearance though creatinine can be regarded as indirect index of muscle mass. Then, correction for all the possible confounders was considered for table 4 (Multivariable analyses).

Table 2: Disposition index. The confidence interval implies significance. Please re-check.

Done.

Table 3: MISI obesity class III seems to have an extreme outlier. Please check.

Done.

Table 3: Some given p values are not significant, even though they indicate trends. Please omit them.

Done.

Discussion:

CLI shows a positive association with obesity and NAFLD, but an inverse association with prediabetes. Please clarify.

We clarify that the positive association occurs unless and until prediabetes or even overt diabetes occurs. Please see section 3.3 Obesity, prediabetes and other metabolic abnormalities and particularly lines 222-224. From the discussion: “Therefore, given the inverse association between clearance during OGTT and prediabetes, we speculate that reduced clearance is key factor in the pathogenesis and occurrence of prediabetes progressing toward overt diabetes.”

Line 206: Pivovarova

Edited.

Further evaluation of the discussion depends on major revision of the aforementioned points.

We believe you will be satisfied reading the revised draft. Otherwise, we will be glad to address further points that may arise.

Overall: The terms "increase" or "decrease" describe processes, changes. In this cross-sectional study, no such changes can be described.

We edited accordingly the text.

Comments on the Quality of English Language few changes needed.

The whole text has been edited.

3.Furthermore, did the authors confirm if their study population met the assumptions for normality in the analyses? In this study, I believe that using generalized estimating equations (GEE) may be more suitable than Univariable and multivariable linear regression analyses.

We agree linear regression was not the best choice. Despite the quite large sample size, there was lack of normality in most of the variables. Hence, we used quantile regression around median instead.

4.Moreover, did the authors conduct Pearson correlation analyses on the data before data mining? If not, I recommend conducting these analyses initially to confirm any existing correlations among the data variables before proceeding with regression analyses.

Done. We enclose in the present draft figure 1 showing matrix of correlations.

Incorporating these revisions and clarifications into the manuscript should enhance its overall quality and understanding for readers.

We agree that comments were very useful to improve quality of the manuscript and we are grateful to this reviewer.

Round 2

Reviewer 2 Report

I have no other opinion.

Reviewer 3 Report

Thanks for the thorough revision. The paper is ready for acceptance.